# Tixagevimab and Cilgavimab (Evusheld™) Prophylaxis Prevents Breakthrough COVID-19 Infections in Immunosuppressed Population: 6-Month Prospective Study

**DOI:** 10.3390/vaccines11020350

**Published:** 2023-02-03

**Authors:** Dejan Jakimovski, Svetlana P. Eckert, Omid Mirmosayyeb, Sangharsha Thapa, Penny Pennington, David Hojnacki, Bianca Weinstock-Guttman

**Affiliations:** 1Jacobs Multiple Sclerosis Center for Treatment and Research (JMSCTR), Department of Neurology, Jacobs School of Medicine and Biomedical Sciences, University at Buffalo, State University of New York, Buffalo, NY 14202, USA; 2Buffalo Neuroimaging Analysis Center (BNAC), Department of Neurology, Jacobs School of Medicine and Biomedical Sciences, University at Buffalo, State University of New York, Buffalo, NY 14203, USA

**Keywords:** persons with neuroinflammatory diseases, COVID-19, breakthrough infection, Omicron, tixagevimab and cilgavimab, Evusheld

## Abstract

Background: Persons with neuroinflammatory diseases (pwNID) treated with potent immunosuppressives are at risk of severe COVID-19 outcomes and reduced vaccine seroconversion. We aimed at determining the real-world efficacy of tixagevimab and cilgavimab (Evusheld™) in immunosuppressed pwNID in preventing breakthrough COVID-19 infections. Methods: 31 immunosuppressed pwNID were followed for 6 months after administration of tixagevimab and cilgavimab as a prophylactic COVID-19 medication (January 2022–July 2022). Only pwNID treated with anti-CD20 monoclonal antibodies and sphingosine-1-phosphate modulators were considered eligible for the study. A control group of 126 immunosuppressed pwNID (38 seropositive and 88 seronegative after SARS-CoV-2 vaccination) were included. Breakthrough COVID-19 infections rate and their severity was determined over the follow-up. Results: The pwNID treated with tixagevimab and cilgavimab had more comorbidities when compared with the total and seronegative pwNID control group (54.8% vs. 30.2% vs. 27.3%, *p* = 0.02 and *p* = 0.005, respectively). After a 6-month follow-up, significantly lower numbers of pwNID treated with tixagevimab and cilgavimab had breakthrough COVID-19 when compared with the control pwNID group (6.5% vs. 34.1%, *p* = 0.002) and seronegative control pwNID group (6.5% vs. 38.6%, *p* < 0.001). All COVID-19 infections in Evusheld-treated pwNID were mild, whereas 9/43 COVID-19 infections in the control group were moderate/severe. No side effects to tixagevimab and cilgavimab were recorded. Conclusion: In pwNID treated with immunosuppressive therapies, tixagevimab and cilgavimab (Evusheld™) significantly reduced the numbers and severity of breakthrough COVID-19 infections during the Omicron (BA.2–BA.5 variants) wave.

## 1. Introduction

In early December 2019, the first severe acute respiratory syndrome caused by a novel enveloped RNA β-coronavirus (SARS-CoV-2) was recorded in Wuhan, China [1]. After rather stable evolutionary stasis in the first 11 months, subsequent SARS-CoV-2 evolution resulted in the emergence of multiple variants that changed the transmissibility and antigenicity characteristics [2]. As of the end of 2022, there have been more than 654 million COVID-19 cases and more than 6.65 million COVID-19-related deaths. Immunosuppressed patients, such as people with malignancies and autoimmune diseases, and solid-organ transplant recipients, are particularly vulnerable to severe outcomes in viral pandemics such as COVID-19 [3,4]. Although most vulnerable populations adopted preventive practices and isolation, the group of patients that utilize therapy which requires hospital access had significantly greater rates of COVID-19 infection [5].

During the peak of the COVID-19 pandemic, people with neuroinflammatory diseases (pwNID) had on average more than double the rate of both severe outcomes (hospitalization and use of ventilator) and mortality [6]. Use of high-dose corticosteroids within 1 month of the infection and use of highly immunosuppressive therapies such as anti-CD20 monoclonal antibodies (mAbs) increased the risk for a worse COVID-19 outcome [6]. Later reports from the second and third wave of COVID-19 concluded that the higher hospitalization and mortality rates may not be associated with use of all immunosuppressive therapies, and the majority of the risk was attributed to demographic well-known risk factors (male sex, older age) [7]. However, the early signals of worse outcomes in patients treated with rituximab remained present in the literature well into the second year of the pandemic [8,9].

The highly immunosuppressive maintenance therapies that are routinely utilized in pwNID significantly decrease the ability to surmount sufficient vaccine-induced seroconversion and provide long-lasting immunity [10,11]. The most common medications used in these populations that are linked with lower seroconversion are the anti-CD20 monoclonal antibodies (mAbs) and the sphingosine-1-phosphate inhibitors [12,13]. On average, only 20% to 30% of PwNID treated with these medications develop any measurable antibody response, and are at higher risk for breakthrough COVID-19 infections [14]. In combination with the lack of seroconversion, some pwNID can also experience both local and systemic adverse reactions to repeated SARS-CoV-2 vaccine administration [15]. Therefore, having the access to an alternative passive immunoprophylaxis medication could significantly decrease the rate and severity of COVID-19 in this vulnerable population. 

One such medication is called tixagevimab and cilgavimab (AZD7442) that consists of two non-overlapping, fully human SARS-CoV-2 neutralizing antibodies with an estimated half-life of 90 days. When compared with placebo, the first pivotal phase 3 trial (PROVENT, NCT04625725) demonstrated that this intramuscular medication provided 82.8% relative risk reduction of symptomatic COVID-19 at a median of 6-month follow-up [16]. No significant differences in adverse and serious adverse events between tixagevimab and cilgavimab combination and placebo were noted [16]. An additional phase 3 trial regarding the use of tixagevimab and cilgavimab as early outpatient treatment of COVID-19 (TACKLE, NCT04723394), demonstrated a significant 45% reduction in severe outcomes or death when compared with placebo [17]. A recent review of the tixagevimab and cilgavimab literature did describe the significant utility of this medication in prevention of breakthrough COVID-19 infections [18]. Up to 38 real-world studies have been performed and showed the significant ability of tixagevimab and cilgavimab to prevent breakthrough infections during the Omicron surge (BA.2) [18]. 

Based on this background, we aimed at determining the safety and efficacy of tixagevimab and cilgavimab in preventing severe breakthrough COVID-19 infections in immunosuppressed pwNID over a 6-month follow-up. These pwNID treated with tixagevimab and cilgavimab were further compared with two control groups of untreated and vaccinated pwNID and to seronegative untreated pwNID populations. We hypothesized that tixagevimab and cilgavimab would significantly decrease the rate and severity of breakthrough COVID-19 infections over the 6-month period. 

## 2. Materials and Methods

### 2.1. Study Population

The subjects were eligible to participate in the study if they had at least one neurologist-confirmed NID (persons with multiple sclerosis (pwMS)) or another inflammatory neurological disease, including clinically isolated syndrome (CIS), radiologically isolated syndrome (RIS), neuromyelitis optica spectrum disorders (NMOSD), myelin oligodendrocyte glycoprotein antibody-associated syndrome (MOG), autoimmune encephalitis (AIE), neurosarcoidosis, and CNS vasculitis). In order for the pwNID to be clinically selected for tixagevimab and cilgavimab treatment, they should have been treated for at least 6 months on one of the B-cell depleting monoclonal antibody (anti-CD20 mAb) therapies (rituximab, ocrelizumab, or ofatumumab) or one of the sphingosine-1-phosphate modulators (S1P; fingolimod, siponimod, ozanimod, or ponesimod). The eligibility was determined by the treating neurologists based on the lack of seroconversion after SARS-CoV-2 vaccination or due to concerns regarding a higher risk of unfavorable COVID-19 outcomes. The exclusion criteria for the pwNID were: (1) being pregnant or a nursing mother, and (2) not willing to perform the study procedures (perform antibody testing before and after tixagevimab and cilgavimab administration). The control pwNID group was based on a previously published study that investigated the effect of disease-modifying therapies on vaccine seroconversion and were treated with either anti-CD20 mAb or S1P modulators.

The presence of breakthrough COVID-19 infections was determined through search of the electronic medical records at the 6-month follow-up timepoint. Additional information regarding the severity of the breakthrough COVID-19 infection and use of COVID-19 specific medications were collected. All COVID-19 infections were separated into mild (presence of catarrhal symptoms, fever or flu-like symptoms without the need for hospitalization), moderate (hospitalization due to COVID-19) and severe (use of oxygen supplementation, placement on ventilators, or mortality). The anti-SARS-CoV-2 antibody titer was determined using FDA-emergency authorized assays such as the Luminescence-based VITROS anti-SARS-CoV-2 IgG test (VITROS ECi/ECiQ/3600 Immunodiagnostic Systems, Rochester, NY, USA) and anti-SARS-CoV-2 Semi-Quantitative Total Antibody 164,090 (LabCorp, Burlington, NC, USA). PwNID treated with tixagevimab and cilgavimab had their anti-SARS-CoV-2 antibody checked before administration of the therapy and at their 6-month visit. All control pwNID had baseline anti-SARS-CoV-2 antibody testing. Seroconversion was determined based on each individual test and the manufacturer-suggested cut-offs. Additional information regarding the use of disease-modifying therapy (DMT), and presence of comorbidities was collected through the electronic medical records. All study participants signed a consent form for data sharing and the study was approved by the University at Buffalo Institutional Review Board (IRB). 

### 2.2. Statistical Analyses

All statistical analyses were performed in SPSS version 26.0 (IBM, Armonk, NY, USA). The data were further visualized in GraphPad version 8.0 (San Diego, CA, USA). Data distribution was determined using visual inspection of the histograms and Q–Q plots. Comparisons using categorical variables was performed by the chi-square test, numerical variables by the Student’s t-test for normally distributed variables and the Mann–Whitney U-test for non-parametric variables. The longitudinal changes in anti-SARS-CoV-2 antibodies in the tixagevimab and cilgavimab group were analyzed using the Wilcoxon signed-rank test. Post-hoc power analysis was performed and the power of the study determined [19], and *p*-values lower than 0.05 were considered as statistically significant. 

## 3. Results

### 3.1. Demographic Characteristics

The demographic and clinical characteristics of the study population are shown in Table 1. Overall, 31 pwNID were treated with tixagevimab and cilgavimab and 126 pwNID served as a vaccinated control group that was not treated with tixagevimab and cilgavimab. There were no age and gender differences between the cohorts. PwNID treated with tixagevimab and cilgavimab had a significantly greater rate of comorbidities when compared with the total control pwNID population (54.8% vs. 30.2%, *p* = 0.02). Moreover, there were no differences in the type of SARS-CoV-2 vaccine used between the groups (*p* > 0.05). The pwNID treated with tixagevimab and cilgavimab had a numerically lower rate of seroconversion when compared with the control group (16.1% vs. 30.2%, *p* = 0.117). When the pwNID population treated with tixagevimab and cilgavimab was compared with only the seronegative pwNID control population, similar differences were noted (*p* = 0.005 for comorbidities and non-significant for other demographic characteristics). None of the pwNID treated with tixagevimab and cilgavimab reported any serious adverse events.

### 3.2. Breakthrough COVID-19 Infections in the Study Populations

The rate of breakthrough COVID-19 infections and severity of the disease are shown in Table 2. Over the 6-month follow-up period, the pwNID treated with tixagevimab and cilgavimab experienced significantly lower rates of COVID-19 breakthrough infections when compared with the total control pwNID group and the seronegative pwNID group (6.5% vs. 34.1% vs. 38.6%, *p* = 0.002 and *p* < 0.001). More importantly, all breakthrough infections in the pwNID treated with tixagevimab and cilgavimab were labeled as mild. Contrarily, numerically larger numbers of control and seronegative pwNID populations with breakthrough COVID-19 infections were labeled as moderate or mild (20.9%, *p* = 0.117 and 6.8%, *p* = 0.338). None of the two pwNID with breakthrough COVID-19 infections previously treated with tixagevimab and cilgavimab required any COVID-19 specific treatment. On the other hand, 20 out of the 43 (46.5%) control pwNID had their COVID-19 infection treated with specific medication (six pwNID on anti-SARS-CoV-2 antibody infusion, seven on nirmatrelvir/ritonavir, one on remdesivir and baricitinib and six pwNID on non-specified medication). 

### 3.3. Longitudinal Antibody Changes in pwNID Treated with Tixagevimab and Cilgavimab

None of the 31 pwNID that received tixagevimab and cilgavimab reported any adverse events to the medication. Only four pwNID had measurable anti-SARS-CoV-2 antibodies before the administration of tixagevimab and cilgavimab. After the administration of the medication, 27 out of 31 (87.1) pwNID had measurable anti-SARS-CoV-2 antibody levels. Twelve pwNID had levels at about the highest measurable limit (index >150) and an average anti-IgG index of 109.8 (median index of 124.9). (longitudinal change in anti-SARS-CoV-2 index is shown in Figure 1) The white blood count and absolute lymphocyte levels were not associated with the amount of anti-SARS-CoV-2 levels after tixagevimab and cilgavimab administration.

## 4. Discussion

The results from this 6-month prospective study demonstrated that treatment with tixagevimab and cilgavimab (Evusheld™) in immunosuppressed pwNID with low seroconversion after vaccination significantly decreased the rate and severity of new breakthrough COVID-19 infections during the period of Omicron BA.2–BA.5 subvariants. As expected, the administration of tixagevimab and cilgavimab (Evusheld™) resulted in an increase of anti-COVID-19 antibody titer that remained sustained over the 6-month period as shown in the sentinal tixagevimab and cilgavimab trials in people without any immunosuppressive characteristics.

One key question regarding the prolonged utility and efficacy of tixagevimab and cilgavimab is the emergence of Omicron variants and sublineages. Mutations and antigenic shift within the glycoprotein (BA.4/BA.5 have additional 69–70 deletion of L452R from the Delta strain, F486V and reversion of the original Q493) [20] have the potential to significantly alter the efficacy of therapeutic monoclonal antibodies. One of the first reports that investigated the effectiveness of prophylactic anti-COVID-19 medications on the ability to prevent breakthrough infections showed lower efficacy against newly emerging SARS-CoV-2 strains (early 2022 and emergence of BA.2). When compared with the Delta strain, the neutralizing ability of casirivimab and imdevimab and/or efficacy of tixagevimab was reduced 344-fold against BA.1 and 9-fold against BA.2 [21]. Additional longitudinal analysis performed a comparison between six different therapeutic mAbs that were tested against Omicron/Delta, BA.2, BA.4 and BA.5; tixagevimab and cilgavimab did neutralized the latest variants (BA.2 and BA.5), albeit with a lower titer and slow decay [22]. Interestingly, the absence in BA.2 and BA.5 of a receptor-binding domain (RBD) from the G446S mutation that is crucial for cilgavimab-neutralizing activity resulted in regaining and increasing the activity against these new SARS-CoV-2 strains [23,24]. Potential stand-alone use of cilgavimab could be hypothesized. On the other hand, the combination of imdevimab and casirivimab (Ronapreve/REGEN-COV) failed to neutralize any of the variants [22]. During the Omicron wave, tixagevimab and cilgavimab pre-exposure prophylaxis in immunosuppressed individuals, such as allogenic stem-cell or solid-organ transplant receipts, resulted in triple or double reduction in COVID-19 incidence when compared with untreated comparators [25,26,27,28]. Moreover, immunocompromised individuals treated with tixagevimab and cilgavimab were 92% less likely to be hospitalized or die when compared with the untreated group [26]. Similar findings were seen in patients with hematological malignancies where patients treated with tixagevimab and cilgavimab were less likely to have breakthrough infections (4.9% vs. 22% in untreated patients) [29]. A recent study showed no differences in the rate of breakthrough COVID-19 infections between patients with B-cell malignancies who were treated with the smaller 150mg dose or the larger 300 mg dose of tixagevimab and cilgavimab [30]. That said, the occurrence of such infections despite vaccination and prophylactic therapy remains a clinically significant concern [30]. In a propensity-matching study of immunocompromised patients, the use of tixagevimab and cilgavimab significantly reduced the risk of COVID-19 by 25% and hospitalization by 59% when compared with the control group [31]. These beneficial effects were better in non-obese patients [31]. A full, recently published list of all regulatory and observational studies regarding the efficacy of tixagevimab and cilgavimab on the reduction rate in hospitalization, ICU use and all-cause mortality are shown elsewhere [18]. 

Only one previous study investigated the use of tixagevimab and cilgavimab in pwMS without any long-term follow-up and a control group [32]. A total of 18 pwMS (17 treated with ocrelizumab and 1 on ofatumumab) were treated with tixagevimab and cilgavimab, and their antibody titer significantly increased over a 2-week period after the administration [3]. At baseline, up to 66% of pwMS did not have detectable antibodies. After tixagevimab and cilgavimab treatment, all 18 subjects had measurable antibodies that were above the highest level of the assay [32]. Similar findings have been seen in other autoimmune diseases that utilize the same anti-CD20 mAbs medications. Tixagevimab and cilgavimab administered in 43 persons with rheumatologic diseases (rheumatoid arthritis, ANCA vasculitis, immune-mediated myositis, Sjögren’s disease, and systemic lupus erythematosus) that were previously treated with rituximab significantly decreased the rate of COVID-19 infections [33]. Only one patient (2.2%) contracted COVID-19 when compared with the 4.32% incidence rate in the local community [34]. Although our study and the rheumatology-based report share the same geographical and socio-economic characteristics, they followed their patients treated with tixagevimab and cilgavimab for half of the recommended duration (3 months vs. 6 months in our study) [33]. A slightly higher rate of breakthrough COVID-19 infection (3 out of 20 cases, 15%) was recorded in the rituximab-treated population with ANCA-associated vasculitis that received tixagevimab and cilgavimab [34].

Some reports in the literature have reported potential cardiovascular adverse events during the use of tixagevimab and cilgavimab [35]. A meta-analysis that utilized published and unpublished data from four randomized clinical trials of tixagevimab and cilgavimab suggested that patients treated with this medication had around 90% greater odds of serious cardiac and vascular adverse events (myocardial infarction, thrombosis or heart failure) [35]. These events were highly likely related to underlying baseline cardiovascular risk factors and no such signal was seen with the use of the higher 600mg dose in the TACKLE trial [36]. Important to mention, the trial that investigated the post-exposure use of tixagevimab and cilgavimab during symptomatic COVID-19 infection (PICO trial, NCT04501978) failed to improve the primary outcome of time to sustained recovery [37]. However, the tixagevimab and cilgavimab group had a significantly lower 30% mortality rate when compared with placebo [37]. 

Some limitations related to this study should be considered. The sample size of pwNID treated with tixagevimab and cilgavimab was relatively small. However, we performed a post hoc power analysis with an expected rate of breakthrough infections in the tixagevimab and cilgavimab group of 5%, and 30% for the untreated control group. In order to have 80% power, the study would have required 24 treated and 96 control pwNID. We exceeded this sample size and are confident in the ability to detect significant and meaningful results. While our study used the follow-up time that was utilized in the seminal tixagevimab and cilgavimab trial in healthy controls, additional long-term effects should also be considered. An additional limitation involving comparison within the study is the change in the recommended dosing of tixagevimab and cilgavimab in the middle of our follow-up period. On 24 February 2022, the FDA revised the recommendation to change the dosing from 150 mg of tixagevimab and 150mg of cilgavimab to 300 mg of tixagevimab and 300 mg of cilgavimab. These changes were the result of the lower neutralizing ability of the medication on the emerging Omicron variants (BA.1–BA.5). In our sample size, only 10 out of the 31 pwNID returned for the additional 150mg doses. That said, we did not have the ability to determine the specific SARS-CoV-2 variate in any of the breakthrough infections. The Omicron subvariants mentioned in this study are based on general prevalence reports issued by the US health authorities. Very recent updates in the Fact Sheet for tixagevimab and cilgavimab have suggested that virus variants with spike substitutions R346T or K444T in combination with F486S or F586V are resistant to neutralization. Variants with such substitutions include the BA.5.26, BF.7, BF.11, BJ.1, BN.1 and XBB types of SARS-CoV-2 virus.

As a response to the decreasing efficacy of tixagevimab and cilgavimab on the newer SARS-CoV-2 strains, the pharmaceutical industry has already started development and testing of adapted monoclonal antibodies. For example, the manufacturers of tixagevimab and cilgavimab are currently conducting a new sentinal Phase I/III trial of testing AZD5156 when compared with both placebo and the Evusheld^TM^ antibodies (SUPERONOVA trial, NCT05648110). The manufactures of such antibody medications are also currently developing production platforms that could easily adjust the targeted antigen and adapt the medication to the currently prevalent viral strain. Such technologies would better equip the health care providers with up-to-date medication that should have efficacy at any given timepoint. An additional trial investigating the use and safety of anti-SARS-CoV-2 antibodies in pregnant patients (NCT05281601) is currently ongoing. Given the move towards newer formulations, the outcome of currently ongoing trials such as the use of 300mg/300mg tixagevimab and cilgavimab in immunosuppressed patients (ENDURE trial, NCT05281601) remains unknown. Other observational and interventional trials regarding the use of tixagevimab and cilgavimab have been reported in clinicaltrials.gov such as the PrEP trial (NCT05461378), COVIMAB (NCT05439044), PRECOVIM (NCT05216588), and EVOLVE (NCT05315323), and are still ongoing, while others such as TIXCI-TRANS (NCT05234398) have been terminated. 

In conclusion, pwNID that fail to mount anti-COVID-19 immune responses after SARS-CoV-2 vaccination would benefit from tixagevimab and cilgavimab (Evusheld™) in prevention of new breakthrough infections and severe COVID-19 outcomes. Prophylactic use of tixagevimab and cilgavimab should be considered in immunosuppressed subjects that are at high risk for severe COVID-19 outcomes. This type of protection can be utilized in pwNID treated with anti-CD20 mAbs or S1P modulators without the need to determine their seroconversion status.

## Figures and Tables

**Figure 1 vaccines-11-00350-f001:**
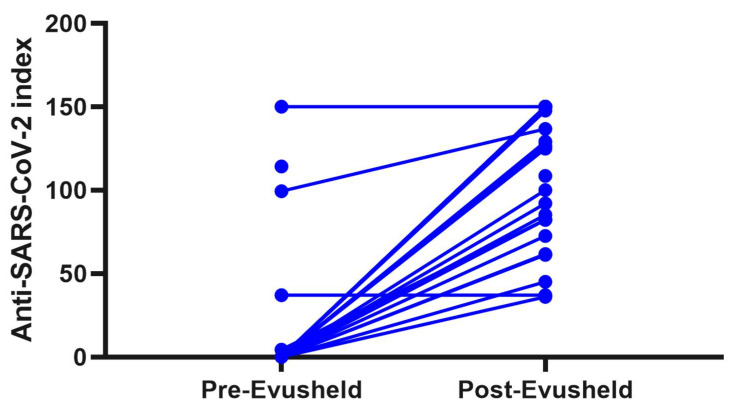
Line plot demonstrating the longitudinal change in anti-SARS-CoV-2 antibodies in pwNID before and after administration of tixagevimab and cilgavimab. Legend: pwNID—patients with neuroinflammatory disorders, a significant response in anti-SARS-CoV-2 antibody levels before and after administration of tixagevimab and cilgavimab in 31 pwNID.

**Table 1 vaccines-11-00350-t001:** Demographic and clinical characteristics of the pwNID study groups.

Demographic and Clinical Characteristics	Evusheld™ Group * (*n* = 31)	Total Control Group (*n* = 126)	Seropositive Control Group (*n* = 38)	Seronegative Control Group (*n* = 88)	Evusheld™ vs. All Controls *p*-Value	Evusheld™ vs. Seronegative Controls *p*-Value
Female, *n* (%)	19 (61.3)	86 (68.3)	26 (68.4)	60 (68.2)	0.545	0.488
Age, mean (SD)	55.1 (10.3)	51.2 (12.4)	49.5 (13.9)	51.9 (11.7)	0.141	0.303
Cardiovascular comorbidities, *n* (%)	17 (54.8)	38 (30.2)	14 (36.8)	24 (27.3)	**0.02**	**0.005**
SARS-CoV-2 vaccine						
BNT162b, *n* (%)	16 (51.6)	67 (53.2)	17 (44.7)	50 (56.8)	0.533	0.444
mRNA 1273, *n* (%)	12 (38.7)	50 (39.7)	21 (55.3)	29 (32.9)		
Ad26.COV2.S, *n* (%)	1 (3.2)	9 (8.1)	-	9 (10.3)		
Seroconversion, *n* (%)	5 (16.1)	38 (30.2)	38 (100)	-	0.117	-
DMT at the time of vaccination						
Anti-CD20 mAbs, *n* (%)	24 (80.0)	103 (81.7)	30 (78.9)	73 (82.9)	**<0.001**	**0.002**
S1Ps, *n* (%)	2 (6.7)	23 (18.3)	8 (21.1)	15 (17.1)		
No DMT	4 (13.3)	-	-	-		

Legend: pwNID—patients with neuroinflammatory disorders, SD—standard deviation, DMT—disease-modifying therapy, S1P—sphingosine-1-phosphate, IQR—interquartile range. * Two pwNID in the Evusheld group were not vaccinated with any SARS-CoV-2 vaccine. One pwNID in the Evusheld group was treated with tocilizumab. *p*-values lower than 0.05 were considered statistically significant and are shown in bold.

**Table 2 vaccines-11-00350-t002:** The effectiveness of tixagevimab and cilgavimab in preventing COVID-19 breakthrough infections and decreasing their severity in pwNID.

Post-Evusheld™ COVID-19 Breakthrough Infections	Evusheld™ Group * (*n* = 31)	Total Control Group (*n* = 126)	Seropositive Control Group (*n* = 38)	Seronegative Control Group (*n* = 88)	Evusheld™ vs. All Controls *p*-Value	Evusheld™ vs. Seronegative Controls *p*-Value
Breakthrough infection, *n* (%)	2 (6.5)	43 (34.1)	9 (23.7)	34 (38.6)	**0.002**	**<0.001**
Moderate/Severe COVID-19, *n* (%)	0 (0)	9 (20.9)	3 (7.9)	6 (6.8)	0.207	0.338

Legend: pwNID—patients with neuroinflammatory disorders. * Two pwNID in the Evusheld group were not vaccinated with any SARS-CoV-2 vaccine. One pwNID in the Evusheld group was treated with tocilizumab. *p*-values lower than 0.05 were considered statistically significant and are shown in bold.

## Data Availability

The data presented in this study are available upon a reasonable request from the corresponding author.

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
