# Peer review of "Tixagevimab and Cilgavimab (Evusheld™) Prophylaxis Prevents Breakthrough COVID-19 Infections in Immunosuppressed Population: 6-Month Prospective Study"

_vaccines, 2023, doi:10.3390/vaccines11020350_

Round 1
Reviewer 1 Report
Dear Authors,
It was a pleasure to read and review the original manuscript “Tixagevimab and cilgavimab (EvusheldTM) prophylaxis prevents breakthrough COVID-19 infections in immunosuppressed population: 6-month prospective study.” Indeed, metabolomics represent a major topic of interest in prevention of a breakthrough of COVID-19 infection in immunosuppressed population. This is one of the first studies to evaluate the effect of tixagevimab and cilgavimab in the rate and severity of breakthrough COVID-19 infection over a 6-month period.
Major Comments:
1. An extremely valuable review was just published in viruses entitled “Tixagevimab/Cilgavimab in SARS-CoV-2 Prophylaxis and Therapy: A Comprehensive Review of Clinical Experience” by Akinosoglou et al. It would be highly recommended to read it and add some relevant information in your introduction/discussion.
2. I would appreciate if you could separate your table 1 into 2 different tables. One table with the general demographic and clinical characteristics of the patients and another one, that you should mention your statistical results.
3. You could also create a figure regarding your follow-up results
Minor Comments:
1) Correct threated line 64
2) Add a space line 65
3) Correct into eligibility – line 98
4) What do you mean by ER ? line 105
5) Correct infeciton line 160
6) TOTAL OF Line 207 (MODIFY THIS PHRASE)
7) Additional limitation that limits the comparison – Improve the phrase Line 240-241
8) BREAK-THROUGH (REMOVE THE - )
2) Major English Editing is required. There are several grammatical issues that must be fixed.
To Sum up,
The paper seems original with a considerable research interest in the recent literature in the field of COVID-19 prevention. It provides information and inspires thoughts in the researched area. It was a pleasure to read, and the readership is going to benefit.
Author Response
Reviewer 1:
Dear Authors,
It was a pleasure to read and review the original manuscript “Tixagevimab and cilgavimab (EvusheldTM) prophylaxis prevents breakthrough COVID-19 infections in immunosuppressed population: 6-month prospective study.” Indeed, metabolomics represent a major topic of interest in prevention of a breakthrough of COVID-19 infection in immunosuppressed population. This is one of the first studies to evaluate the effect of tixagevimab and cilgavimab in the rate and severity of breakthrough COVID-19 infection over a 6-month period.
Response: We thank the Reviewer for the positive comments and the comprehensive review. It significantly improved the quality of our manuscript. Point-by-point response are shown hereafter.
Major Comments:
- An extremely valuable review was just published in viruses entitled “Tixagevimab/Cilgavimab in SARS-CoV-2 Prophylaxis and Therapy: A Comprehensive Review of Clinical Experience” by Akinosoglou et al. It would be highly recommended to read it and add some relevant information in your introduction/discussion.
Response: We thank the Reviewer for the excellent suggestion. We have utilized the recommended review and supplemented our Introduction. This review of the literature is very timely and significantly improved our manuscript.
- I would appreciate if you could separate your table 1 into 2 different tables. One table with the general demographic and clinical characteristics of the patients and another one, that you should mention your statistical results.
Response: The single large table has been separated in two tables. The Table 1 contains only the baseline demographic and clinical results, whereas Table 2 demonstrates the breakthrough COVID-19 related information.
- You could also create a figure regarding your follow-up results
Response: We thank the Reviewer for the suggestion. We have included a line-plot figure that demonstrates the significant response in anti-SARS-CoV-2 antibodies after the administration of tixagevimab and cilgavimab.
Minor Comments:
1) Correct threated line 64
Response: This has been corrected.
2) Add a space line 65
Response: This has been corrected.
3) Correct into eligibility – line 98
Response: This has been corrected.
4) What do you mean by ER ? line 105
Response: This has been corrected.
5) Correct infeciton line 160
Response: This has been corrected.
6) TOTAL OF Line 207 (MODIFY THIS PHRASE)
Response: This has been corrected.
7) Additional limitation that limits the comparison – Improve the phrase Line 240-241
Response: This has been corrected.
8) BREAK-THROUGH (REMOVE THE - )
Response: This has been corrected.
2) Major English Editing is required. There are several grammatical issues that must be fixed.
Response: We have run the manuscript through grammar software and corrected all grammatical and typo mistakes.
To Sum up,
The paper seems original with a considerable research interest in the recent literature in the field of COVID-19 prevention. It provides information and inspires thoughts in the researched area. It was a pleasure to read, and the readership is going to benefit.
Response: We thank the Reviewer for the positive comments.

Reviewer 2 Report
First of all, congratulations to the authors for their work. The study is interesting and provides information that is of interest from the point of view of protecting vulnerable populations. However, I believe that the study suffers from some important limitations that may weaken the conclusions reached by the authors.
One is the number of cases studied, which seems to me to be too few for the conclusions reached. Did the authors calculate the power of the sample?
On the other hand, the study was carried out at a time when the new variants against which EVUSHELD seems to have little efficacy were not in circulation and this and this may make the results less valid.
Regardless of the above comments, the article is well written. The introduction adequately reviews the topic and puts it in context. The objectives are well defined and the methodology is appropriate. The results are understandable and the discussion is correct.
Only one comment in the discussion in relation to the sentence: "Administration of tixagevimab and cilgavimab (Evusheld™) resulted with increase of antiCOVID-19 antibody titer that remained sustained over the 6-month period", this has not been an objective of the study so I do not think the comment is relevant.
Author Response
Review 2:
First of all, congratulations to the authors for their work. The study is interesting and provides information that is of interest from the point of view of protecting vulnerable populations. However, I believe that the study suffers from some important limitations that may weaken the conclusions reached by the authors.
Response: We thank the Reviewer for the positive comments and the comprehensive review. It significantly improved the quality of our manuscript. Point-by-point response are shown hereafter.
One is the number of cases studied, which seems to me to be too few for the conclusions reached. Did the authors calculate the power of the sample?
Response: The Reviewer is correct. We have performed a post-hoc power analysis regarding our study. Using the expected breakthrough infection rate in the treated patients of 5% and 30% as expected rate within our control group (the actual rate was 34.1%) we have calculated the power analysis with 4:1 ratio. Based on these criteria, the required sample size to determine 80% power was 24 (for treated pwNID) and 96 (for control group). This has been included in the manuscript and further commented upon. The calculation of the power analysis is shown in the attached file.
On the other hand, the study was carried out at a time when the new variants against which EVUSHELD seems to have little efficacy were not in circulation and this and this may make the results less valid.
Response: We agree with the Reviewer. We have further explained the time frame of the study and the lack of efficacy on the current strains of the virus. Moreover, we have now expanded on the upcoming new monoclonal antibody that has better efficacy on the new strains. (AZD5156 in the SUPERNOVA trial; NCT05648110)
Regardless of the above comments, the article is well written. The introduction adequately reviews the topic and puts it in context. The objectives are well defined and the methodology is appropriate. The results are understandable and the discussion is correct.
Only one comment in the discussion in relation to the sntence: "Administration of tixagevimab and cilgavimab (Evusheld™) resulted with increase of antiCOVID-19 antibody titer that remained sustained over the 6-month period", this has not been an objective of the study so I do not think the comment is relevant.
Response: We have corrected and explained the sentence. Due to comments from other Reviewers in this manuscript, we have decided to keep this sentence and further explain it.

Reviewer 3 Report
The SARS-CoV_2 pandemic has left humanity with experiences that should be learned from to prevent future catastrophic events. There are numerous publications about the efficacy and safety of texagevimab and cilgavimab to prevent COVID-19 or decrease its severety effect (Alhumaid et al., 2022). Some of these studies are based on a systematic review and meta-analysis of other researcher´ papers or data banks. Unfortunately, case-studies are more than limited. The present manuscript is a solid, correctly designed and applied , 6-month prospective analysis of the immunosuppresed population reaction to prophylaxis treatment. The concrete example of original study of persons with neuroinflammatory deseases (pwNID) is discussed. The manuscript is well-written and easy to understand by interested researchers or, why not, the public in general. Notwithstanding the mechanism of serum neutralization of Omicron sublineages is not mentioned. To overpass the feeling that it is just a correctly reported case study with comprehensive corresponding references discussion. The recent work of Bruel et al. published in Nature Medicine (June, 2022), whose other paper (Ref. 18) cited by the Authors, as well as the excelent work of Touret et al. (2022, Nature Scientific Reports) about in vitro evaluation of therapeutic antibodies against SARS-CoV-2 Omicron B.1.1.1.529 isolate, can be helpful.
Author Response
Reviewer 3:
The SARS-CoV_2 pandemic has left humanity with experiences that should be learned from to prevent future catastrophic events. There are numerous publications about the efficacy and safety of texagevimab and cilgavimab to prevent COVID-19 or decrease its severety effect (Alhumaid et al., 2022). Some of these studies are based on a systematic review and meta-analysis of other researcher´ papers or data banks. Unfortunately, case-studies are more than limited. The present manuscript is a solid, correctly designed and applied , 6-month prospective analysis of the immunosuppresed population reaction to prophylaxis treatment. The concrete example of original study of persons with neuroinflammatory deseases (pwNID) is discussed. The manuscript is well-written and easy to understand by interested researchers or, why not, the public in general. Notwithstanding the mechanism of serum neutralization of Omicron sublineages is not mentioned.
Response: We thank the Reviewer for the positive comments and the comprehensive review. It significantly improved the quality of our manuscript. Point-by-point response are shown hereafter. We have also implemented the Alhumaid et al. study that was mentioned in the comment.
To overpass the feeling that it is just a correctly reported case study with comprehensive corresponding references discussion. The recent work of Bruel et al. published in Nature Medicine (June, 2022), whose other paper (Ref. 18) cited by the Authors, as well as the excelent work of Touret et al. (2022, Nature Scientific Reports) about in vitro evaluation of therapeutic antibodies against SARS-CoV-2 Omicron B.1.1.1.529 isolate, can be helpful.
Response: We have significantly expanded our Discission of the manuscript by expanding on the work from Bruel et al and Touret et al. We have further discussed the efficacy on different strains and the upcoming new similarly designed medications such as the AZD5156 in the SUPERNOVA trial (NCT05648110)
